# Associations between Taste and Smell Sensitivity, Preference and Quality of Life in Healthy Aging—The NutriAct Family Study Examinations (NFSE) Cohort

**DOI:** 10.3390/nu14061141

**Published:** 2022-03-08

**Authors:** Shirley X. L. Lim, Richard Höchenberger, Niko A. Busch, Manuela Bergmann, Kathrin Ohla

**Affiliations:** 1Cognitive Neuroscience (INM-3), Institute of Neuroscience and Medicine, Research Center Jülich, Wilhelm-Johnen-Straße, 52428 Jülich, Germany; 2NutriAct-Competence Cluster Nutrition Research Berlin-Potsdam, 14558 Nuthetal, Germany; bergmann@dife.de (M.B.); kathrin.ohla@gmail.com (K.O.); 3CEA, Inria, Université Paris-Saclay, 1 Rue Honoré d’Estienne d’Orves, 91120 Palaiseau, France; richard.hoechenberger@gmail.com; 4Institute of Psychology, University of Münster, Fliednerstraße 21, 48149 Münster, Germany; niko.busch@wwu.de; 5German Institute of Human Nutrition Potsdam-Rehbrücke, Arthur-Scheunert-Allee 114-116, 14558 Nuthetal, Germany; 6Experimental Psychology Unit, Helmut Schmidt University/University of the Armed Forces Hamburg, Holstenhofweg 85, 22043 Hamburg, Germany; 7Firmenich SA, Rue de la Bergère 7, 1242 Satigny, Switzerland

**Keywords:** taste, smell, Quality of Life, sensitivity, threshold, QUEST

## Abstract

Taste and smell function decline with age, with robust impairment in the very old. Much less is known about taste and smell function in young and middle aged. We investigated taste and smell sensitivity via thresholds in a sub-sample of the NutriAct Family Study (NFS), the NFS Examinations cohort (NFSE; *N* = 251, age *M* = 62.5 years). We examined different aspects relating to taste and smell function: the degree to which taste and smell sensitivity relate to another and to taste and smell preferences, the role of gender and age, as well as effects on Quality of Life (QoL). Taste thresholds were highly correlated, but no correlation was observed between taste and smell thresholds and between thresholds and preference. Women were more sensitive for both taste and smell than men. We found no effect of age on sensitivity and no effect of sensitivity on QoL. All null findings were complemented by Bayesian statistics. Together our results indicate the independence of taste and smell despite their overlap during sensorial experiences. We found no evidence for age-related sensory decline, which could be due to our sample’s characteristics of non-clinical volunteers with good dental health and 93% non-smokers.

## 1. Introduction

The ability to taste and smell are important determinants of food perception and appreciation. Smell additionally contributes to the detection of potentially threatening stimuli such as the smell of spoiled food, fire, or pollution, and it helps kin recognition (see [1]). Similar to sight and hearing, taste and smell decline with age as a result of sensory senescence [2], with robust impairment occurring in the very old [3,4]. For example, in the NHANES cohort, smell impairment has been shown to increase with age: from 4.2% for ages 40–49, over 12.7% for ages 60–69, up to 39.4% for age 80+ years [5]. Although age-related sensory decline is evident, there is no consensus on the prevalence of taste and smell dysfunction at different ages or at which age significant taste and smell impairments are to be expected in the population. This can, at least in part, be attributed to different populations investigated (e.g., clinical, healthy, etc.), different measurements or techniques used (e.g., self-assessment, validated tests, etc.), and other factors than age influence chemosensory function (e.g., smoking, comorbidities, overweight, etc.).

Unlike for sight and hearing, no corrective means such as lenses or hearing aids exist to compensate for diminished taste and smell, making secondary complications in areas relating to taste and smell more likely. For example, taste and smell impairment have been shown to affect numerous aspects of everyday life leading to reduced appetite [6], mood changes and depression [7,8], and reduced Quality of Life, particularly in clinical populations (QoL; [9,10]).

Much less is known about taste and smell function in healthy young and middle old, which were in the focus of the present study. For this, we examined taste and smell in a sub-sample of the NutriAct Family Study on Determinants of Food Choice (NFS; [11]), the NutriAct Family Study Examinations cohort (NFSE; *N* = 251, age: *M* = 62.5 years). The NFS/E are non-clinical cohorts of volunteers who were invited to participate in a longitudinal study protocol. Because this is the first report on the NFSE cohort, we will describe its complete study protocol, although the present study focuses only on different aspects that have previously been linked with taste and smell function: the degree to which taste and smell sensitivity relate to taste and smell preferences, respectively, the role of age, gender, smoking, and dental health, as well as effects of taste and smell on QoL. We also examined the extent to which thresholds for different tastes align with each other to assess whether sensitivity for either taste would permit inference about sensitivity to other tastes and hence could be considered a marker of general taste sensitivity. Lastly, we assessed the association between taste and smell sensitivity.

## 2. Methods

### 2.1. Recruitment

An eligible sub-sample of the NFS [11] was invited to participate in the NutriAct Family Study Examinations (NFSE), an on-site study involving a series of physical and cognitive examinations (see Figure 1, for an overview of the study protocol). Eligibility for invitation required that index persons were at least 50 years old and that at least two family members (e.g., a spouse and a sibling) agreed to participate as well. Eligible persons were mailed an invitation letter, a brochure, and a reply form with a prepaid return envelope. Interested families were then contacted by phone to arrange appointments. A total of 374 participants were invited to the human study center of the German Institute of Human Nutrition; 251 participants took part between August 2018 and September 2019. Participants gave written informed consent before participation. The study protocol conformed with the revised Declaration of Helsinki and was approved by the ethical board of the Landesärztekammer Brandenburg (reference: S 21(a)/2015).

### 2.2. NFSE Protocol

#### 2.2.1. NFS Data Transfer

Eating behavior was assessed with the German version of the Dutch Eating Behavior Questionnaire (DEBQ; [12]), QoL with the Short Form-8 Health Survey (SF-8; [13]), and educational status was categorized according to the Comparative Analysis of Social Mobility in Industrial Nations Project (CASMIN) classification [14]. These data were collected online within the NFS and transferred to the NFSE. The average time between participation in the NFS and the NFSE was 18.1 ± 4.63 months.

#### 2.2.2. Pre-Visit Survey

Prior to the visit at the study center, participants were asked to complete a web-based survey with brief check-box questions to assess the ability to taste and smell, menopausal status (women only), chronic diseases, surgeries or injuries of the mouth, nose, ears, weight changes, and smoking of tobacco products. Additionally, they completed the Trier Inventory for Chronic Stress (TICS; [15]).

#### 2.2.3. Examination Procedure

The visit at the study center lasted up to 4 h. During this time, participants completed the following measurements, tests, and tasks in a fixed order with short breaks in between tests as needed (see Figure 1B). First, blood pressure was taken and the self-reported hunger (7-point rating scale anchored with 1 = no hunger and 7 = very hungry) was rated. A total of 3 participants indicated they were hungry (hunger score > 4) and received a banana before they continued. To determine whether a participant could safely partake in all tests and examinations, they answered health questions including the self-report of medication. For instance, the bioimpedance analysis (BIA) to measure body composition could not be conducted when the participant had electrical devices or large metallic parts implanted; or the Bogus Taste Test would have to be adapted for participants reporting lactose intolerance. Next, three computerized tasks were completed: an Approach–Avoidance Task (AAT; [16]), a Dot Probe Task [17], and a Go–No Go Task [18]. Then, a Bogus Taste Test (BGTT; [19]) with apples, carrots, chocolate, and salted peanuts was completed. Next, anthropometric measures-weight, height, circumferences of waist and hip, and body composition using BIA were taken. Then, participants performed a heart rate variability (HRV; [20]) and heart beat tracking task (HBDT; [21]). After this, blood and hair samples were collected to be stored in the biobank. Next, participants completed a computer-based interview on the dietary intake off the day before the visit (24-h recall). Then, hand grip strength was measured. Then, they rated alertness on a 7-point scale with 1 = tired and 7 = very awake and again hunger before they rated how much they could eat of their favorite dish now (on a 7-point scale from 1 = nothing to 7 = as much as I could get). Finally, taste and smell sensitivity [22,23] were measured.

At the end of the visit, participants were equipped with an accelerometer (ActiGraph wGT3X-BT) to record physical activity for 8 days. They also received a kit for stool sample collection including a prepaid return-envelope. During the following 12 months after the visit, participants were contacted via phone for three 24 h dietary recalls roughly 3 months apart. The calls were made on different weekdays to reduce the risk that participants foresaw the call and adapted their diet accordingly.

### 2.3. Taste and Smell Sensitivity

#### 2.3.1. Questionnaires

DEBQ, CASMIN, and FS-8 data were collected during the NFS. The German version of the DEBQ [12] measures three dimensions of eating style with 30 items: the degree of restrained, emotional, and external eating. Education was assessed according to the internationally comparable CASMIN-index [24] as previously reported for the NFS cohort [25]. The SF-8 measures the eight health profile dimensions of the SF-36 [26] in a comparatively short test using one item per dimension: physical functioning, bodily pain, role limitations due to physical health problems, role limitations due to personal or emotional problems, emotional well-being, social functioning, energy/fatigue, and general health perceptions. Participants evaluate each item on a 5- or 6-point Likert scale, which is then standardized according to the scoring system, where weights are applied to each item [13]. The four items representing physical and psychological dimensions are summed and represent the physical (PCS-8) and mental score (MCS-8), respectively.

Before threshold measurement, participants were asked to rate how much they liked (preference) sour, salty, sweet, and bitter foods and beverages as well as sweet, spicy, smoked and citrus smells on separate 5-point scales anchored with “not at all” and “very much”.

#### 2.3.2. Taste Threshold Measurement

Tastants were prepared by diluting prototypical chemicals that are known to elicit a clear taste perception in deionized water: citric acid (sour; *M* = 192.12 g/mol), sodium chloride (salty; *M* = 58.44 g/mol), quinine hydrochloride dihydrate (bitter; *M* = 396.91 g/mol), and sucrose (sweet; *M* = 342.30 g/mol). Concentrations were equidistantly spaced on a decadic logarithmic grid for each tastant based on an established protocol [23,27]: citric acid, 0.015 mM to 46.846 mM (14 log_10_ steps; step width: 0.269); sodium chloride, 0.342 mM to 342.231 mM (12 log_10_ steps; step width: 0.273); quinine, 0.077 × 10^−3^ mM to 3.131 mM (21 log_10_ steps; step width: 0.23); and sucrose, 0.073 mM to 584.283 mM (14 log_10_ steps; step width: 0.3). Taste solutions were presented at room temperature. Each taste stimulus was presented manually to the anterior half of the tongue using a conventional glass bottle with a spray head. Aliquots were approximately 0.2 mL. Thresholds are henceforth referred to as by their taste quality (sweet, sour, etc.).

Participants were blindfolded during the taste test to minimize distraction and improve focus. At the beginning of each threshold run, they were told which taste would be tested next. Before presenting individual taste samples, they were asked to stick out the tongue and expect the receipt of the stimulus. Participants were to indicate whether they recognized the taste by nodding (“yes”) or shaking their head (“no”) without moving their tongue in. Participants rinsed their mouths with deionized water after a response. The maximum number of trials per threshold was 20. We used an adaptive test algorithm based on QUEST that has been shown to be reliable and quick [23,27,28]. The interval between any two taste stimuli was approx. 30 s. Participants received no feedback as to their performance during the experiment.

Thresholds estimated above highest stimulus concentration (that is outside the stimulus range) would systematically underestimate sensitivity. To counter this, we adjusted the threshold to the highest stimulus concentration in 15 cases (4 citric acid, 2 sodium chloride, 2 sucrose, and 7 quinine). Visual inspection of the raw data revealed that in very few cases participants responded (almost) exclusively “Yes” to any stimulus. Given the wide concentration ranges including miniscule concentration, it is extremely unlikely that a participant recognizes all stimuli. This is further corroborated by prior observations in younger samples, where no participant could recognize all tastes. Hence, we deemed thresholds implausible if they contained ≤2 “no” responses and removed them from analysis. This was the case for 45 taste thresholds in 29 participants. Of these, all four thresholds were excluded in four participants. Taste thresholds were not recorded (missing) for three participants. Overall, 912 taste thresholds are reported here (222 bitter, 233 sweet, 227 sour, and 230 salty).

#### 2.3.3. Statistical Analyses

Differences in threshold estimates between tastes were assessed with a one-factorial repeated measures analysis of variance (rmANOVA) with the between-subjects factor gender (women and men) with four levels (sour, salty, sweet, and bitter). Greenhouse–Geisser correction for violation of sphericity was applied (Mauchly’s test of sphericity). *p*-values were Bonferroni-corrected for multiple comparisons and indicated as *p*_bonf_. Uncorrected *p*-values are reported. The alpha-level was set to 0.05 for all statistical tests.

Spearman correlations were computed to explore the relation between the four taste thresholds and between taste composite and smell threshold. Independent pairwise comparisons were performed with Welch’s Test whenever assumptions of equal variance were violated (Levane’s Test). Linear regression analyses were used to test whether the taste threshold predicts the respective taste preference (sour, salty, sweet, or bitter) and whether the smell threshold predicts respective smell preference (sweet, spicy, smoked, or citrus). Linear mixed models were used to investigate age effects (fixed effects) on taste and smell sensitivity using gender as random intercept and slope.

Along with conventional NHST, Bayesian analyses were performed to complement Linear Regressions and Linear Mixed Models and to support interpretation of potential null findings. Bayes Factors (*BF*) were calculated to indicate how likely the observed data are under the alternative hypothesis *H*_1_ relative to the null hypothesis *H*_0_. The *BF* quantifies the relative predictive performance of the two rival hypotheses. Although there are no strict bounds for the *BF*, there are conventions that help guide *BF* classification [29]. Accordingly, we interpret the evidence for *H*_0_ as extreme for *BF*_10_ < 0.01, very strong for 0.01–0.03, strong for 0.03–0.1, moderate for 0.1–0.3. Evidence for either hypothesis is considered anecdotal for *BF*_10_ ranging from 0.3 to 3. Evidence for *H*_1_ is considered moderate for *BF*_10_ 3–10, strong for 10–30, very strong for 30–100, and extreme for values >100.

For Bayesian independent *t*-tests we used a Cauchy prior width of 0.707. For Bayesian Linear Regressions we used a “medium” scaling factor on the JSZ-prior (*r* scale = 0.354), a uniform model for prior distribution, and the MCMC sampling method (no. samples = 10,000).

#### 2.3.4. Software

Stimulus presentation and data collection were guided by a Python computer program based on PsychoPy 1.85.4 on Windows 7 (Microsoft Corp., Redmond, WA, USA). Statistical analyses were performed with Jamovi 1.6.23 [30]. Linear regressions and linear mixed models were implemented with GAMLj [31] in Jamovi. Bayesian statistics were computed with the BAS package [32,33] within Jamovi. Figures were plotted in Python using matplotlib v2.2.2 [34] and seaborn v0.11.1 [35].

## 3. Results

### 3.1. Participants

A total of 251 volunteers participated in the study. Five participants did not complete the taste/smell threshold measurement; their datasets were removed from the sample. One participant identified as neither male nor female and was removed from the sample because most analyses included a binary gender comparison. Data of the remaining 245 participants (138 women; 107 men; age in years: *M* = 62.5, *SD* = 5.23, range: 50–81) are presented here. Their age and gender distribution are shown in Figure 2.

As summarized in Table 1, participants were slightly overweight (BMI *M =* 25.6). Participants had good dental health with 84.1% of participants reporting ≤4 deafferented or extracted teeth in the lower jaw and the vast majority (93.1%) were non-smokers. They reported eating behavior in the DEBQ that corresponds well with normative data [12]. Participants had a high secondary to low tertiary education according to the CASMIN classification [14]. The observed QoL reports were well within the range reported in a German sample [36] for men; women reported higher QoL for both mental (MCS-8) and physical (PCS-8) dimensions, than would be expected from reference data [36].

### 3.2. Taste and Smell Sensitivity

As expected, thresholds differed significantly between tastes (rmANOVA: *F*_2.69,559.87_ = 1309.46, *p* < 0.001, *η*_p_^2^ = 0.863; Figure 3A), with sweet (*M* = 1.28log_10_ mmol/L, *SD* = 0.58) yielding the highest thresholds, followed by salty (*M* = 0.57log_10_ mmol/L, *SD* = 0.57), sour (*M* = 0.097, *SD* = 0.62) and bitter (*M* = −1.16, *SD* = 0.79). Women generally had lower taste thresholds than men (*F*_1,208_ = 12.1; *p* < 0.001, *η*_p_^2^ = 0.055), yet when resolving the interaction between taste and gender (*F*_2_._69,559.87_ = 3.14; *p* = 0.025, *η*_p_^2^ = 0.015), significantly lower thresholds in women compared to men (post hoc pairwise comparisons, *N* = 4) were confirmed for sour (*t*_208_ = 3.412, 95% CI = [0.0204, 0.543], *p*_bonf_ < 0.01) and bitter (*t*_208_ = 3.294, 95% CI = [0.0136, 0.6811], *p*_bonf_ < 0.01) but not for salty (*t*_208_ = 1.994, 95% CI = [−0.0822, 0.362], *p*_bonf_ = 0.194) and sweet (*t*_208_ = 1.471, 95% CI = [−0.124, 0.341], *p*_bonf_ = 0.571). The smell thresholds for PEA (Figure 3C) were on average −0.876log mmol/L (*SD* = 1.09, min = −3.28, max = 1.32), corresponding to pen number 6.09 (*SD* = 3.44, min = 1, max = 13.9).

### 3.3. Link between Taste and Smell Sensitivity

Spearman correlation analysis revealed strong correlations between the thresholds for the four tastes (*rho* from 0.46 to 0.55; Table 2). Given the correspondence between tastes, we computed a composite score for taste sensitivity (Figure 3B). For this, we averaged the z-transformed thresholds for each taste, which were computed across participants. This composite score was then used for further analyses. In contrast, no significant correlation was found between any of the taste thresholds and the smell threshold for PEA (Table 2).

### 3.4. Sensitivity and Gender

Following up on the observed gender effect for the different taste thresholds (Figure 3A), we assessed the magnitude of the gender difference for the taste composite threshold. As expected, we found that the taste composite thresholds were significantly lower for women (*M* = −0.05, *SD* = 0.63) than men (*M* = 0.28, *SD* = 0.79; unpaired Welch’s *t*-test: *t*_168_ = 3.29, *p* = 0.001, *d*= 0.464). This was corroborated by strong evidence for a gender difference (*BF*_10_ = 29.9). Similarly, smell thresholds (Figure 3C) were significantly lower in women (*M* = −1.13, *SD* = 1.06) compared to men (*M* = −0.56, *SD* = 1.06; Welch’s *t*-test: *t*_224_ = 4.15, *p* < 0.001, *d* = 0.54), which was also corroborated by decisive evidence for a gender difference (*BF*_10_ = 188.08).

### 3.5. Sensitivity and Preference

Linear regression analyses showed that neither of the four taste thresholds predicted the corresponding taste preference (sweet: ${\hat{R}}^{2}$ = −0.004, β_1_ = 0.008, *p* = 0.832; sour: ${\hat{R}}^{2}$ = 0.006, β_1_ = −0.069, *p* = 0.123; salty: ${\hat{R}}^{2}$ = 0.0000342, β_1_ = 0.034, *p* = 0.338; bitter: ${\hat{R}}^{2}$ = −0.002, β_1_ = 0.004, *p* = 0.421; Figure 4A). Those findings were corroborated by moderate evidence for *H*_0_ (sweet: *BF*_10_ = 0.157, salty: *BF*_10_ = 0.179, sour: *BF*_10_ = 0.207, bitter: *BF*_10_ = 0.171).

Similar results were obtained for smell sensitivity, which did not predict smell preference for either of the four smell qualities assessed (sweet: ${\hat{R}}^{2}$ = −0.015, β_1_ = 0.148, *p*_bonf_ = 0.136; smoked: ${\hat{R}}^{2}$ = 0.005, β_1_ = 0.097, *p*_bonf_ = 1; citrus: ${\hat{R}}^{2}$ = −0.002, β_1_ = 0.058, *p*_bonf_ = 0.52; spiced: ${\hat{R}}^{2}$ = −0.001, β_1_ = −0.068, *p*_bonf_ = 1; Figure 4B). Those findings were corroborated by anecdotal evidence for *H*_1_ for sweet (*BF*_10_ = 1.2), anecdotal evidence for H_0_ for smoked (*BF*_10_ = 0.419), and moderate evidence for *H*_0_ for spiced (*BF*_10_ = 0.198) and citrus (*BF*_10_ = 0.179).

### 3.6. Sensitivity and Age

To test whether age contributes to taste sensitivity, we next fitted a linear mixed model to predict the composite taste threshold with age (Figure 5A). Given the gender effect, we included gender as a random effect with varying slopes and intercepts. The model’s total explanatory power was insubstantial (conditional *R*^2^ = 0.0934) and the part corresponding to age alone was 0.0000164 (marginal *R*^2^). The model’s intercept (β_0_) is 0.149 (95% CI = [−1.065, 1.3630.462], *t*_93.9_ = 0.241, *p* = 0.810). Within this model, the effect of age was not significant (β_age_ = −0.000564, 95% CI = [−0.0192, 0.0181], *t*_207_._7_ = −0.0592, *p* = 0.953). In line, the *BF*s provided no evidence for an age effect in women (*BF*_10_ = 0.21) and in men (*BF*_10_ = 0.29). Separate models for each of the taste qualities yielded similar findings (see Appendix A).

We then fitted a linear mixed model to predict smell sensitivity with age including gender as a random effect (Figure 5B). The model’s total explanatory power was insubstantial (conditional *R*^2^ = 0.110) and the part corresponding to age alone is 0.00381 (marginal *R*^2^). The model’s intercept (β_0_) is 0.0239 (95% CI = [−0.456, 0.504], *t*_0_._989_ = −0.098, *p* = 0.938). Within this model, the effect of age was not significant (β_age_ = 0.0121, 95% CI = [−0.0123, 0.0365], *t*_237_._91_ = 0.975, *p* = 0.331). In line, the *BF*s provided anecdotal evidence for *H*_0_ in women (*BF*_10_ = 0.50) and in men (*BF*_10_ = 0.24).

### 3.7. Sensitivity and Quality of Life (QoL)

We first tested whether QoL differed as a function of gender. We found no significant gender effect for neither the mental (Welch’s *t*-test: *t*_221_ = 0.601, *p* = 0.584, *d* = 0.078) nor the physical (Welch’s *t*-test: *t*_225_ = 0.238, *p* = 0.812, *d* = 0.031) facet of QoL. These findings were supported by strong evidence for H_0_ (physical: *BF*_10_ = 0.15; mental: *BF*_10_ = 0.17). Mental and physical QoL scores were 52.8 (mental: range 17.2–60.1, *SD* = 7.63) and 49.5 (physical: range 26.0–61.1, *SD* = 8.45) for men and 52.2 (mental: range 17.2–63, *SD* = 7.17) and 49.2 (physical: range 24.3–60.8, *SD* = 8.19) for women, respectively.

Consequently, we used simple linear models to test whether taste and smell sensitivity predicted QoL. Specifically, we fitted separate linear models for taste (Figure 6A) and smell sensitivity (Figure 6B) with the mental and physical facets of QoL as dependent variables. Neither the mental nor the physical QoL was significantly predicted by taste (mental: ${\hat{R}}^{2}$ = −0.006, *p* = 0.133; physical: ${\hat{R}}^{2}$ = 0.012, *p* = 0.115) or smell sensitivity (mental: ${\hat{R}}^{2}$ = −0.002, *p* = 0.23; physical: ${\hat{R}}^{2}$ = 0.001, *p* = 0.267; see Table 1 for complete statistics). Those null findings were supported by anecdotal evidence for *H*_0_ for taste (physical: *BF*_10_ = 0.49; mental: *BF*_10_ = 0.44) and moderate evidence for *H*_0_ for smell (physical: *BF*_10_ = 0.25; mental: *BF*_10_ = 0.28).

To complement our findings, we computed four separate linear regressions for each of the four taste qualities (see Appendix A). As for the taste score, we found no effect of taste sensitivity for either of the four tastes on the mental facet of QoL. However, the physical facet of QoL could be predicted by sweet (*p* = 0.04) and sour (*p* = 0.03) taste sensitivity.

## 4. Discussion

### 4.1. Taste and Smell Sensitivity

Taste thresholds were strongly correlated indicating that each taste threshold represents a good proxy of taste sensitivity in general. Our finding is surprising with respect to the strengths of the observed taste-taste associations because the magnitude of previous taste-taste correlations varied markedly between studies and taste qualities–spanning from no correlations at all [37], even between substances of the same taste qualities [38], over weak or modest [39,40,41] to strong [42] correlations. Part of this variability between studies can probably be explained by differences in the types of threshold (e.g., recognition versus detection), stimulation method (e.g., whole mouth versus part of the tongue), the granularity or precision of thresholds, and the demographics of the sample (e.g., age, gender, or education).

In contrast to the taste–taste correlations, we found near zero correlations between taste and smell sensitivity. This finding is in line with a previous report advocating the statistical independence of chemosensory sensitivities [42] despite significant overlap in the cortical processing of the chemical senses smell, taste, and chemesthesis [43].

### 4.2. Sensitivity and Preference

While taste is undoubtedly a crucial factor in food choices [44], the role of sensitivity for different tastes in taste preference is debated. Most studies that found a link between sensitivity and preference focused on bitter taste compounds with a strong genetic component such as PROP for which a heightened bitter sensitivity is typically accompanied by reduced preference for bitter foods [45]. Numerous studies indicate that PROP taster status is associated with generally heightened taste sensitivity [46,47] but associations with preference varied between studies and showed enhanced preference [48], reduced preference [49], and no effect on preferences [44,50], depending on the taste quality. The current body of evidence makes it thus difficult to predict a directed link between taste sensitivity and preference.

We observed not only no link between the sensitivity and the self-reported preference for the four basic taste qualities tested here, but we found moderate evidence in favor of the null hypothesis that postulates that taste sensitivity is not associated with the self-reported preference for foods of the respective taste quality. The clear evidence provided by our data may be owing to the large sample size. It is in line with other observations that taste sensitivity is not related to food liking/consumption [51,52], which together with our findings suggest that sensitivity and preference for a given taste substance can be related, but this link does not translate to the preference of foods.

Similar to taste, we observed no evidence for an association of the rose threshold with preference for any of the four different smell qualities. We even found moderate evidence against such an association for spiced and citrus but not for sweet and smoked smells. This variability could be due to inter-individual variation in the understanding of the smell qualities given that smells are notoriously difficult to identify and, vice versa, the smell labels subjects were given may have been difficult to consistently link to real-world objects. Admittedly, we cannot exclude that the smell qualities rated for preference may poorly relate to the rose-like smell used for threshold measurement and other qualities or threshold odors may yield different results.

Together, previous inconsistent findings may be the result of different approaches to preference. The preference for a taste solution or an odorous molecule is not the same as the preference for a food category or smell quality. Alternatively, biases in sampling may have exhibited their effects in particularly small samples. It has been shown that taste and flavor preferences begin to form through exposure early in life [53,54,55] and the authors of a taste study in children [56] even speculated that a “higher preference for and acceptance of sour taste could also foster higher intake of acidic foods and thereby improve the ability to perceive sour taste”. Although numerous factors influence food intake [57], differences in exposure, and with that possibly sensitivity and/or preference, could be dependent on education, socio-economic status, or culture and have a notable impact in small samples.

### 4.3. Sensitivity and Gender

Women have been traditionally suggested to outperform men in chemosensory abilities. A closer look into the bulk of literature, however, reveals that this notion has been derived from conceptionally different measures such as sensitivity thresholds, identification tasks, behavioral reports, or neuroimaging. Some of these measures differ in the required level of cognitive abilities (e.g., memory or language) or subjectivity (e.g., self-report) that may introduce gender differences independent of taste or smell sensitivity. For example, gender differences in nasal trigeminal sensitivity, assessed with ratings and EEG, were independent of sensitivity and interpreted as differences in cognitive appraisal between genders [58].

We found lower smell and taste thresholds (i.e., higher sensitivity) for women compared to men indicating an increased chemosensory sensitivity of women in our sample of middle-aged to old adults. Numerous studies on smell sensitivity have yielded no gender differences for rose odor [59] and also for other odors such as n-butanol [60]. However, whenever gender differences were observed, women were more sensitive than men (see [61], for a review). A recent meta-analysis including 8848 participants found higher olfactory sensitivity in women compared to men with small to medium effect sizes [62], similar to our study where the effect sizes were medium for both taste (Cohen’s *d* = 0.46) and smell (*d* = 0.54), suggesting that some previous reports may not have had the sample size needed to uncover a gender effect.

For taste, much less prior work exists, and studies vary in their test protocols (e.g., taste drops, taste sprays, whole mouth stimulation). Nevertheless, similar to smell, a higher sensitivity of women has been reported. Specifically, lower thresholds [63] or higher taste scores in a taste drop test [64] were found in women compared to men. When comparing young (19–33 years) and older (60–75 years) adults, only older men were less sensitive than young men than women, suggesting that gender-related differences in taste sensitivity are influenced by age and may thus only be observed in samples with an appropriate age-range In addition, gender differences may be specific to some taste qualities and not others as indicated by the report that sour, salty, and bitter but not sweet sensitivity differed between men and women [63]. However, in this study [63], smoking is likely a confounding factor as about 70% of the men and only 7% of the women were smokers.

### 4.4. Sensitivity and Age

We found no age-related decline in taste or smell sensitivity in our sample. This stands in contrast to numerous previous studies reporting age-related decline in olfactory [39,65,66] and nasal trigeminal sensitivity [67]. For example, Hummel and co-workers [65] showed a decrease in overall smell function with age which was most pronounced for smell thresholds but also in smell identification and discrimination in a sample of 3282 subjects aged 5 to >55 years. For taste sensitivity, previous findings are more diverse and include reports of age-related decline [64,68] but also null findings in a sample ages 19–79 [39]. This disaccord may, in part, also reflect the large inter-individual variation in taste perception [69]. Sampling a wider age range including adolescents ([64]; 14 to 79 years) as well as contrasting two different age groups (19–33 vs. 60–75 years) [68] may have been contributors to previous reports of age-related decline.

The bulk of literature identified age-related decline in smell and taste via paired comparisons between different age groups, which may be more sensitive to detect differences than the regression analyses we used here. However, our sample differs from most previous studies in at least two aspects that may further contribute to the seemingly discrepant results. First, our sample has a comparably narrow age range of 50–81 years with most participants aged between 57 and 67 years. If sensory decline occurred with increasing age, it would be most probably detected in a sample that covers a large part of the lifespan as previous studies that covered a much wider age range [39,68], and that even include adolescents [64] or children [65] suggest.

Secondly, some previously published samples include clinical clientele, i.e., patients that visited the clinic and had their taste and smell assessed as part of a medical examination, which increases the likelihood to return a test result that is, in the worst case, confounded by a medical condition. In contrast, participants of the NFSE are a group of interested and mobile volunteers who were able to travel to the study center and partake in numerous tests and exams and complete numerous online surveys. One may suspect a sampling bias of particularly healthy individuals in our study. A comparison of the actual smell threshold values in the NFSE sample with those of previous reports yielded, however, similar results [66,70,71,72], thus making a sampling bias unlikely.

### 4.5. Quality of Life (QoL)

The ability to smell plays a major role in the enjoyment of food, social interactions, but also the detection of threats such as fire or spoiled food. Smell and taste loss may thus exhibit multifarious ramifications on everyday life and have been associated with a reduced QoL. This has been observed in patients with taste disturbances due to cancer [73] and in different patient groups with severe smell disturbances [74]. However, those reports include patients presenting to physicians specialized in smell and taste disorders due to a high disease burden and they are thus not representative of the general population. In fact, although the prevalence for smell loss is estimated to be as high as 19–24%, many patients seem to cope well, likely because they are unaware of their chemosensory deficits [75].

Nevertheless, those with chemosensory impairment show reduced olfactory-related QoL and this effect is exacerbated for self-report compared to objective tests [74], highlighting the role of disease awareness. Here, we find no effect of taste or smell sensitivity on QoL measured with the FS-8. This is not surprising, given that our participants showed on average no evidence for notable smell or taste impairments; quite the opposite was true, their thresholds compared well to those of a younger sample tested with the same protocol [22,27].

## 5. Conclusions

Together our results indicate the function independence of taste and smell despite their overlap during sensorial experiences. Supporting previous findings of heightened chemosensory sensitivity in women, we found no evidence for age-related sensory decline, which could be due to our sample´s characteristics of non-clinical volunteers with good dental health and 93% non-smokers.

## Figures and Tables

**Figure 1 nutrients-14-01141-f001:**
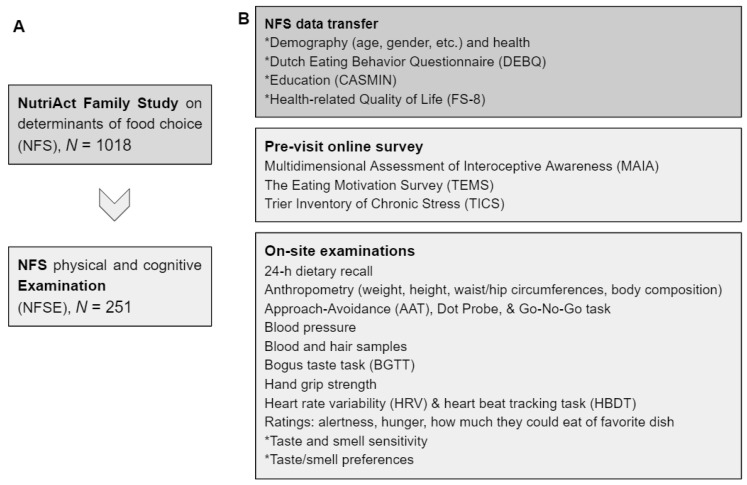
Design of the NFSE-he NutriAct Family Study on determinants of food choice (NFS) physical and cognitive Examinations. (**A**): A sub-sample (*N* = 251) of the NFS participated in the NFSE. (**B**): Data collected from NFSE participants in three phases: during the NFS, during a pre-visit online survey, and during on-site examinations. The data reported here are indicated by a star (*).

**Figure 2 nutrients-14-01141-f002:**
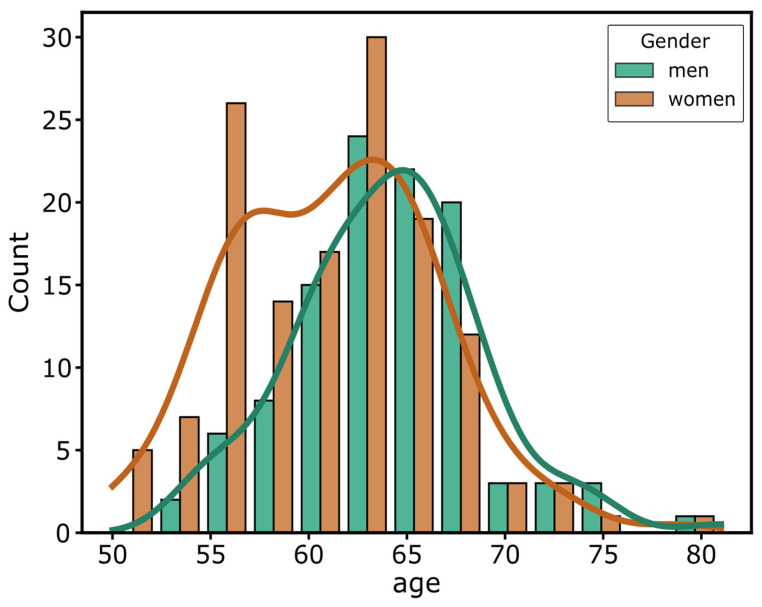
Age distribution of the NFSE sample by gender. Mean (±*SD*) age 62.5 (±5.23) years. Minimum age = 50, maximum age = 81 years. *N* = 245, 138 women and 107 men.

**Figure 3 nutrients-14-01141-f003:**
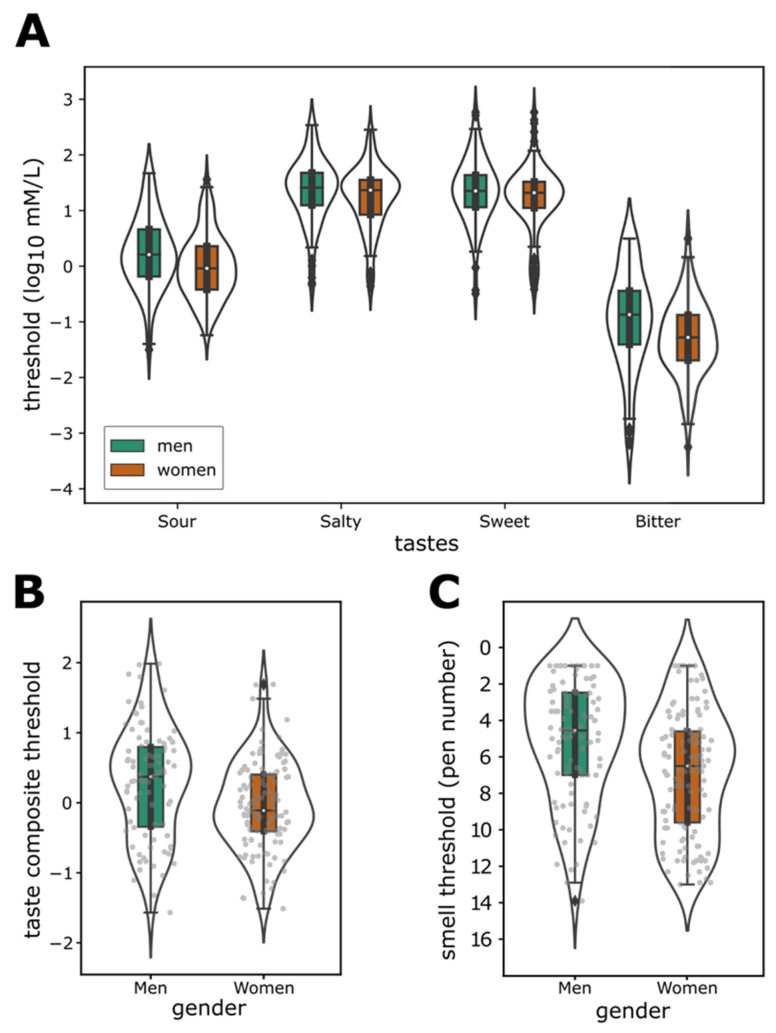
(**A**) Taste threshold estimates for salty, sour, sweet, and bitter. (**B**) Taste composite score (averaged z-transformed of the four tastes) for women and men. (**C**) Smell threshold estimates for PEA (rose-like smell) for women and men. Higher pen numbers indicate lower thresholds. Bars represent Tukey’s boxplot with violin plots superimposed on, whiskers show the upper and lower boundary of 1.5 inter-quartile range, dots are individual data points.

**Figure 4 nutrients-14-01141-f004:**
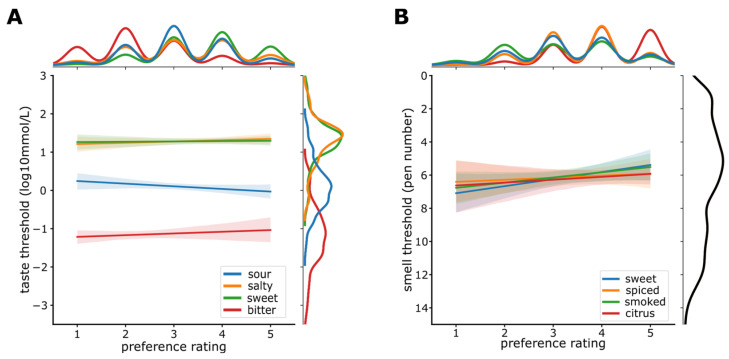
(**A**) Scatter plot with linear fit for taste threshold for different taste qualities and their corresponding taste preference rating. Colors indicate taste qualities. (**B**) Scatter plot with linear fit for rose smell threshold and preference ratings for four odor qualities. Colors indicate odor qualities. Shaded regions show 95% CI (confidence interval). Density functions for preference ratings are plotted at the top and for taste/smell thresholds on the right of each figure.

**Figure 5 nutrients-14-01141-f005:**
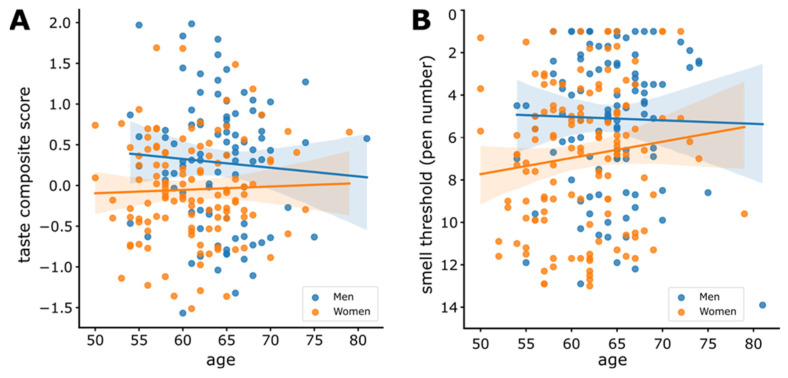
(**A**) Scatter plot with linear regression line of taste composite scores fitted across age. (**B**) Scatter plot with linear regression line of smell thresholds (pen number) fitted across age, for men and women separately. Shaded regions represent 95% CI. Each dot is an individual’s data point with color indicating gender.

**Figure 6 nutrients-14-01141-f006:**
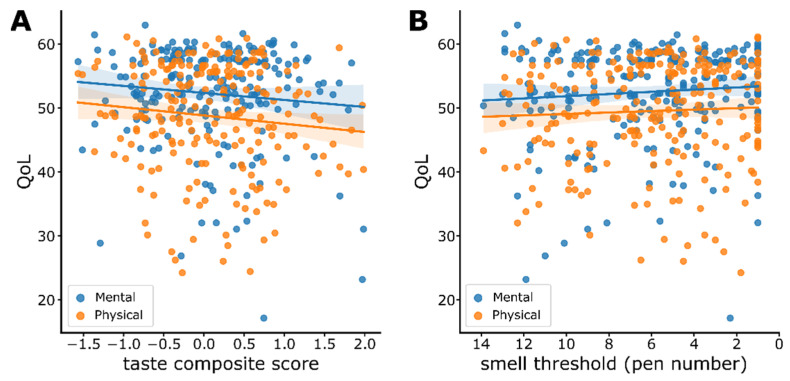
(**A**) Scatter plot to show the relationship between Quality of Life (QoL) and taste composite scores. A linear regression line was fitted separately for each component of QoL (mental and physical). (**B**) Scatter plot to show the relationship between QoL and smell thresholds. A linear regression line was fitted separately for each facet of QoL (mental and physical). Shaded regions show 95% CI. Each dot represents an individual’s data point.

**Table 1 nutrients-14-01141-t001:** Demographics of participants. Means ± Standard deviations.

	Women (*N* = 138)	Men (*N* = 107)	All (*N* = 245)
Age (years)	61.3 ± 5.27	64.1 ± 4.75	62.5 ± 5.23
Body Mass Index (BMI)	25.1 ± 4.3	26.3 ± 3.44	25.6 ± 3.99
*N* smokers	11	6	17
Education level CASMIN *N* (%)			
3	82 (59.42%)	84 (78.50%)	166 (67.76%)
2	54 (39.13%)	20 (18.69%)	74 (30.20%)
1	2 (1.45%)	3 (2.80%)	5 (2.04%)
Eating Behavior (DEBQ)			
Total	2.39 ± 0.47	2.12 ± 0.4	2.27 ± 0.46
Emotional subscale	1.74 ± 0.66	1.42 ± 0.46	1.60 ± 0.60
Restraint subscale	2.95 ± 0.72	2.60 ± 0.70	2.79 ± 0.73
External subscale	2.49 ± 0.50	2.33 ± 0.47	2.42 ± 0.50
Quality of Life (SF-8)			
MCS-8 (mental)	52.8 ± 7.63	52.2 ± 7.17	52.4 ± 7.37
PCS-8 (physical)	49.5 ± 8.45	49.2 ± 8.19	49.3 ± 8.29

**Table 2 nutrients-14-01141-t002:** Spearman correlations between z-transformed taste thresholds for sweet, salty, sour, and bitter.

	Sour		Salty		Bitter		Sweet		Rose	
Sour	-									
Salty	0.520	***	-							
Bitter	0.538	***	0.527	***	-					
Sweet	0.514	***	0.527	***	0.46	***	-			
Rose	0.014		0.049		0.058		−0.002		-	

*** Indicates *p* < 0.001.

## Data Availability

Due to the nature of this prospective research study, the data are pseudonymous and participants of this study did not agree for their data to be shared publicly, so supporting data is not available.

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
