# Peer review of "Associations between Taste and Smell Sensitivity, Preference and Quality of Life in Healthy Aging—The NutriAct Family Study Examinations (NFSE) Cohort"

_nutrients, 2022, doi:10.3390/nu14061141_

Round 1

Reviewer 1 Report

This manuscript report the result of association study between the chemical (taste and olfaction) sensitives and physical conditions (age, gender, and QOL) in relatively older populations called “young and middle old” (50-81yo). Because they selected the participants with healthy conditions from the cohort, the results are not related to the illness. They showed that taste thresholds are highly correlated each other; however, the thresholds between taste and olfaction are not. These thresholds and preferences were not correlated. There is no relationship between the thresholds and age or QOL. Only the correlation between the gender and the thresholds are evident as reported in other studies. Composite thresholds of taste or olfaction are lower in females than males.

The authors showed high correlation between the thresholds of each basic tastes; however, it is necessary to show the results of the original plots between the thresholds of each basic tastes and age as well as QOL, because they are independent basic tastes. Even in the gender, they did not show the significance of the correlation with each basic tastes. In reference 63, the difference between age and genders are reported. The profiles are different between the basic tastes, which allow more deep discussion. There are several related reports with similar or opposite results. The authors should try more detailed discussion, otherwise, the knowledge of the field become wider but not deeper.

Author Response

This manuscript report the result of association study between the chemical (taste and olfaction) sensitives and physical conditions (age, gender, and QOL) in relatively older populations called “young and middle old” (50-81yo). Because they selected the participants with healthy conditions from the cohort, the results are not related to the illness. They showed that taste thresholds are highly correlated each other; however, the thresholds between taste and olfaction are not. These thresholds and preferences were not correlated. There is no relationship between the thresholds and age or QOL. Only the correlation between the gender and the thresholds are evident as reported in other studies. Composite thresholds of taste or olfaction are lower in females than males.

  • The authors showed high correlation between the thresholds of each basic tastes; however, it is necessary to show the results of the original plots between the thresholds of each basic tastes and age as well as QOL, because they are independent basic tastes. Even in the gender, they did not show the significance of the correlation with each basic tastes. 

Reply: The reviewer raises an important point that we are happy to address. We have now computed separate linear models for each of the four taste qualities for the variables age (Supp. Figure 1 A-D) and QoL (Supp. Figure 2 A-D) - men and women are plotted separately to allow the reviewer to evaluate possible gender differences following our plotting conventions in the manuscript. (Please refer to the new Supplementary Material file to inspect the figures and statistics.)

We computed separate regression models for each taste - they yielded very similar results to our regression in the main manuscript where we used the “taste composite score”: there is no significant age effect for any taste quality. We now include the plots and related regression models for each taste as Supplementary Material and reference the results in the main manuscript.  

Similar to the overall taste score, we found no effect of taste sensitivity for either of the four tastes on the mental facet of QoL. However, the physical facet of QoL could be predicted by sweet (p=0.04) and sour (p=0.03) taste sensitivity. We now include the plots and related regression models for each taste as Supplementary Material and reference the results in the main manuscript.  

  • In reference 63, the difference between age and genders are reported. The profiles are different between the basic tastes, which allow more deep discussion. There are several related reports with similar or opposite results. The authors should try more detailed discussion, otherwise, the knowledge of the field become wider but not deeper.

Reply: We absolutely agree and expanded the text in the discussion to account for the taste-specific effects reported in reference 63. The modified text now reads:

“Specifically, lower thresholds [63] or higher taste scores in a taste drop test [64] were found in women compared to men. When comparing young (19-33 years) and older (60-75 years) adults, only older men were less sensitive than young men than women, suggesting that gender-related differences in taste sensitivity are influenced by age and may thus only be observed in samples with an appropriate age-range. In addition, gender differences may be specific to some taste qualities and not others as indicated by the report that sour, salty, and bitter but not sweet sensitivity differed between men and women [63]. However, in this study [63], smoking is likely a confounding factor as about 70% of the men and only 7% of the women were smokers.

Reviewer 2 Report

The author investigates whether the taste and smell sensitivity is associated with their relative preference. In addition, the association between taste and smell sensitivity with other factors such as gender, age, and Quality of life has also been studied. This is an excellent study because of his unique cohort and brilliant discussion. Although most of the associations have been studied before, the current study gives us some contradictory results, leading to an excellent opportunity to discuss the difference between the previous design of studies and the power of evidence. Generally speaking, the cohort used here is most healthy volunteers, mostly between 57 and 67 (relatively young), compared with most previous studies with more aged populations or pre-existing diseases. We have a few suggestions on the paper before it can be finally accepted.

  1. First, as the author suggested in the discussion, it is not surprising to see no association between the smell and taste and the Quality of life in this cohort. However, we wonder if it is appropriate to draw this conclusion based on a cohort with average no evidence for notable smell or taste impairments. It should be stated in the results and discussion, though.
  2. Due to the lack of enough samples with severe impairments on the taste and smell, we wonder if this is also the reason for the lack of significant association in sensitivity and preference. In Parkinson's disease, a patient can have no clinical symptoms even with 20% of the SN dopaminergic loss. This is not because of no association between neuronal loss and clinical symptoms. The observable association could only be found after it reaches a certain threshold. We have the same concern for the conclusion of no age-related decline in taste or smell sensitivity. For most neurodegenerative diseases, aging is 1st predominant factor. Without a cohort including enough aged population, the correlation between age and disease progression won't be found. Instead, genetic mutations become the 1st contributing factor. 
  3. In line 416, the author suggests that the previous reports may not have enough sample size to uncover a gender effect. Is there any power analysis that has been done to support this?
  4. Please keep consistent on the Quality of life (QoL or QOL?)

Author Response

The author investigates whether the taste and smell sensitivity is associated with their relative preference. In addition, the association between taste and smell sensitivity with other factors such as gender, age, and Quality of life has also been studied. This is an excellent study because of his unique cohort and brilliant discussion. Although most of the associations have been studied before, the current study gives us some contradictory results, leading to an excellent opportunity to discuss the difference between the previous design of studies and the power of evidence. Generally speaking, the cohort used here is most healthy volunteers, mostly between 57 and 67 (relatively young), compared with most previous studies with more aged populations or pre-existing diseases. We have a few suggestions on the paper before it can be finally accepted.

  • First, as the author suggested in the discussion, it is not surprising to see no association between the smell and taste and the Quality of life in this cohort. However, we wonder if it is appropriate to draw this conclusion based on a cohort with average no evidence for notable smell or taste impairments. It should be stated in the results and discussion, though.

Reply: We agree with the reviewer that it would not be appropriate to conclude that taste and smell sensitivity are not related to QoL - not based on our data but also not based on the current literature. And we make no such statement in our manuscript. Quite the opposite, we mention previous reports of reduced QoL associated with severe smell/taste impairment in clinical populations. For milder forms of smell/taste impairment, the literature suggests that most patients do not report a reduced “general” QoL. This is also not surprising given that QoL includes many (more important) aspects than the chemical senses. This is also supported by the fact that tests that measure olfactory-related QoL (with a focus on smell experiences) are more sensitive to uncover the negative effects in smell/taste impaired individuals - but these are specific to smell-related situations. One important factor that may contribute to QoL impairments is the awareness of the smell/taste impairment. Accordingly, individuals who have not noticed their smell/taste impairment cannot suffer from it.

The lack of association of smell/taste sensitivity and QoL in our sample is consistent with the literature and very plausible because we observed no major smell/taste impairments.    

On page 13 it reads: “Smell and taste loss may thus exhibit multifarious ramifications on everyday life and have been associated with a reduced QoL. This has been observed in patients with taste disturbances due to cancer [73] and in different patient groups with severe smell disturbances [74]. However, those reports include patients presenting to physicians specialized in smell and taste disorders due to a high disease burden and they are thus not representative of the general population. In fact, although the prevalence for smell loss is estimated to be as high as 19-24%, many patients seem to cope well, likely because they are unaware of their chemosensory deficits [75]. Nevertheless, those with chemosensory impairment show reduced olfactory-related QoL and this effect is exacerbated for self-report compared to objective tests [74], highlighting the role of disease awareness. Here, we find no effect of taste or smell sensitivity on QoL measured with the FS-8. This is not surprising, given that our participants showed on average no evidence for notable smell or taste impairments; quite the opposite was true, their thresholds compared well to those of a younger sample tested with the same protocol [22,27].”

  • Due to the lack of enough samples with severe impairments on the taste and smell, we wonder if this is also the reason for the lack of significant association in sensitivity and preference. In Parkinson's disease, a patient can have no clinical symptoms even with 20% of the SN dopaminergic loss. This is not because of no association between neuronal loss and clinical symptoms. The observable association could only be found after it reaches a certain threshold. We have the same concern for the conclusion of no age-related decline in taste or smell sensitivity. For most neurodegenerative diseases, aging is 1st predominant factor. Without a cohort including enough aged population, the correlation between age and disease progression won't be found. Instead, genetic mutations become the 1st contributing factor. 

Reply: We also agree with the reviewer on this point and we apologize if this was not made clear enough in the paper. Our sample is indeed neither clinical nor very old - it is a group of  “middle” old largely healthy volunteers. So we can not draw any conclusions on clinical or older populations or possible genetic causes.

  • In line 416, the author suggests that the previous reports may not have enough sample size to uncover a gender effect. Is there any power analysis that has been done to support this?

Reply: The reviewer raises an important point - many studies indeed do not report effect sizes nor sufficient data that allow a posteriori computation of effect sizes. It is thus very difficult for us to make a definitive assessment of effect sizes in other studies. However, we found “medium” effect sizes for gender effects in smell and taste in our sample - this is similar to a recent meta-analysis with over 8.000 participants. So, we can assume that a medium effect size of about 0.5 is representative for the gender difference in chemosensory sensitivity.  

If we make an exemplary calculation for a comparison between two independent groups (e.g. men and women) assuming a true effect size of 0.5 given a type II error rate of 0.2, type I error rate (alpha) of 0.05 (two-tailed), one would need N=126 to uncover the effect. While the estimate of the sample size (N) will differ with different statistical models, it provides an idea of the sample size needed to uncover a gender difference in smell/taste sensitivity. Most previous studies report much smaller samples than this. 

  • Please keep consistent on the Quality of life (QoL or QOL?)

Reply: We apologize for the oversight and have corrected this.